# Analyzing Tuberculosis Reactivation in Patients with Rheumatoid Arthritis and Ankylosing Spondylitis Treated with Biological Therapy Using Machine Learning Methods

Andra-Maria Mircea-Vicoveanu [1], Elena Rezuș [2], Florin Leon [3] and Silvia Curteanu [1,*]

1 Faculty of Chemical Engineering and Environmental Protection, "Gheorghe Asachi" Technical University of Iași, 700050 Iași, Romania; andramircea@yahoo.com
2 Department of Rheumatology and Physiotherapy, "Grigore T. Popa" University of Medicine and Pharmacy, 700115 Iași, Romania; elena.rezus@umfiasi.ro
3 Faculty of Automatic Control and Computer Engineering, "Gheorghe Asachi" Technical University of Iași, 700050 Iași, Romania; florin.leon@academic.tuiasi.ro
* Correspondence: silvia_curteanu@yahoo.com

**Abstract:** This study is based on the consideration that the patients with rheumatoid arthritis and ankylosing spondylitis undergoing biological therapy have a higher risk of developing tuberculosis. The QuantiFERON-TB Gold test result was the output of the models and a series of features related to the patients and their treatments were chosen as inputs. A distribution of patients by gender and biological therapy, followed at the time of inclusion in the study, and at the end of the study, is made for both rheumatoid arthritis and ankylosing spondylitis. A series of classification algorithms (random forest, nearest neighbor, k-nearest neighbors, C4.5 decision trees, non-nested generalized exemplars, and support vector machines) and attribute selection algorithms (ReliefF, InfoGain, and correlation-based feature selection) were successfully applied. Useful information was obtained regarding the influence of biological and classical treatments on tuberculosis risk, and most of them agreed with medical studies.

**Keywords:** rheumatoid arthritis; ankylosing spondylitis; tuberculosis; machine learning algorithms; classification

## 1. Introduction

It is known that the patients with rheumatoid arthritis and ankylosing spondylitis undergoing biological therapy have a higher risk of developing tuberculosis (TB). This is the reason why the diagnosis and treatment of latent tuberculosis infections (LTBI) in these patients are very important.

A study conducted in Korea aimed to estimate the risk of TB development among rheumatoid arthritis (RA) patients receiving biological disease-modifying antirheumatic drugs (bDMARDs) or a Janus kinase (JAK) inhibitor. The study included a total of 765 RA patients (where 59 were JAK inhibitor users, 132 were non-TNF inhibitor users, and 574 were TNF inhibitor users), with a positivity rate of 26.5% (*n* = 203) [1].

Another study [2] included 38,702 patients with ankylosing spondylitis (AS), other types of spondyloarthritis (SpA), and psoriatic arthritis (PsA), and 200,417 persons from the general population. Among all patients, 11 active TB cases were identified. The stconcluded that biological-naive patients do not have an increased TB risk, but the risk increases for the patients following the treatment with biological agents.

Machine learning algorithms are being used in medicine in order to determine which patients are at high risk of developing a disease. In a study published in 2020, the random forest algorithm was used to identify latent tuberculosis (LTBI) patients with an active tuberculosis profile. The QuantiFERON-TB Gold Test was used to detect the tuberculosis infection. A class prediction study was conducted using the WEKA software on the train

set. From the three candidate machine learning algorithms (naive bayes, random forest, and sequential minimal optimization), random forest had the best performance in the leave-one-out cross-validation procedure, and it was selected to create a classification model. The model was validated in the second cohort of patients, with an accuracy of 89% correctly classified instances and 89% sensitivity [3].

In a study conducted by Liu Z. [4], 17 patients with early rheumatoid arthritis were evaluated at baseline for gene expression profiles. After a mean of 5 years, their disease status was evaluated using a combined index (pain, and the global and recoded modified health assessment questionnaire (MHAQ) scores). Nineteen "predictor genes" of future disease severity were identified using a supervised t-test analysis. The results were validated in an independent cohort of RA patients using two supervised machine learning algorithms: support vector machines (SVM) and k-nearest-neighbor classification (kNN). The study demonstrated that the peripheral blood gene expression profiles can predict future disease severity in patients with early and established RA.

In another study [5], ribonucleic acid (RNA) from peripheral blood mononuclear cells (PBMCs) of 9 RA patients and 13 healthy volunteers was analyzed on an oligonucleotide array. Twenty-nine transcripts were identified that were preferentially expressed in RA patients and a list of predictor genes was assembled using the k-nearest neighbor method. The study concluded that analysis of RA PBMCs may provide a set of candidate genes that could aid in the early diagnosis and treatment of RA patients.

A study published in 2020 [6], compared the relative efficacy of infliximab (IFX) at 5 mg/kg and intravenous golimumab (GOL IV) at 2 mg/kg in ankylosing spondylitis patients. The nearest neighbor matching algorithm was used to match patients from the GOL IV group with patients from the IFX group. The GOL IV group showed significantly greater improvements in ASAS20 responses than the IFX group for weeks 28 to 44 in AS patients. The GOL IV group was comparable to the IFX group in changes from the baseline Bath Ankylosing Spondylitis Functional Index (BASFI) scores and C-reactive protein (CRP) levels.

In a retrospective study conducted by Joo Y.B. [7], two independent axial spondyloarthritis (axSpA) groups were used as training and testing datasets. They tested the feasibility of supervised machine learning algorithms to predict radiographic progression in axSpA patients. Radiographic progression was identified in 25.3% and 23.7% of patients in each group. Five classifiers were used in order to obtain a consensus-based classification for the phenogroups: a generalized linear model (GLM), decision trees (DT), k-nearest-neighbors (kNN), support vector machines (SVM), and naive bayes (NB). The SVM and GLM were the top two best-performing models. Clinical and radiographic data-driven predictive models performed reasonably well in predicting the radiographic progression of axSpA.

Another study by Castro-Zunti et al. [8] used machine learning algorithms (k-nearest neighbors and random forest) and deep learning-based classifiers to detect erosions on computed tomography (CT) imagery as an early AS symptom. In their study, the random forest classifiers outperformed the kNN classifiers and the best deep learning classifier was the one trained without minimizing validation loss. The results indicate the potential of machine and deep learning to aid the diagnostics of AS based on CT imagery.

The present study aims to analyze the incidence of latent tuberculosis in patients with rheumatoid arthritis and ankylosing spondylitis treated with biological disease-modifying antirheumatic drugs (bDMARDs), depending on the biological agent and the treatment with conventional DMARDs (csDMARDs) and nonsteroidal anti-inflammatory drugs (NSAIDs).

The importance of this analysis lies in the fact that the medical problem has a practical utility, and the training data belong to real patients. The study included 76 patients diagnosed with RA and 63 patients diagnosed with AS, undergoing biological treatment with original and biosimilar bDMARDs, in monotherapy or in combination therapy. Based on the therapeutic protocols regarding the use of biological agents in RA and AS patients,

they were monitored for tuberculosis infections by performing the QuantiFERON TB-Gold test prior to and during the biological treatment. QuantiFERON-TB Gold test results were positive in 21.05% of RA patients and 23.8% of AS patients during the biological therapy, suggesting the presence of a tuberculosis infection.

The classification problem aims to predict an indicator of tuberculosis (the results of the QuantiFERON-TB Gold Test), given a series of statistics about the patients, such as gender, age, the biological agent, the duration of biological therapy, if the therapy is monotherapy or combination therapy, the history of tuberculosis (for the RA patients), the date, and the result of the QuantiFERON-TB Gold test. In order to confirm that these attributes were relevant to our problem, three attribute-selection algorithms were used: ReliefF, correlation-based feature selection (CFS) and information gain (InfoGain).

The datasets were analyzed using machine learning algorithms. For classification, the following algorithms that generally give good results were applied: random forest, k-nearest neighbors (kNN), C4.5 decision trees, and support vector machines (SVM). For each algorithm, the most important parameters were varied, such as the number of trees for random forest, the number of neighbors and the weighting function of the importance of the neighbors for k-nearest neighbors, pruned or unpruned versions for the C4.5 decision trees, and the kernel type for support vector machines.

The results obtained are presented in two variants: on the entire training set to highlight the ability of a model to learn the data, and with a 10-fold cross-validation to highlight the generalization capability of the learned model, which refers to the model's ability to respond to unseen data in the training phase.

Satisfactory results were obtained in the simulation, proving the efficiency of the applied methods.

## 2. Medical Framework

Rheumatoid arthritis (RA) and ankylosing spondylitis (AS) are inflammatory diseases with an important socioeconomic impact, for which early diagnoses and effective treatments are essential. Over 50% of patients with rheumatoid arthritis cease their professional activity in the first 5 years of illness, and approximately 10% of them suffer a severe disability in the first 2 years of the disease. Ankylosing spondylitis begins in the most productive period of life (18–30 years) and has a rapidly progressive evolution to ankylosis, which influences retirement in the first year after the diagnosis in 5% of patients. Disability affects 80% of patients after 10 years [9].

When the first-line pharmacological agents do not cause remission and the patients maintain a high activity of the disease, the treatment of these conditions involves the use of second-line therapies with biological disease-modifying antirheumatic drugs (bDMARDs). They involve the risk of the reactivation of hepatitis B and C virus infections, as well as of latent tuberculosis, which is why patients are evaluated in this regard before initiating therapy with biological DMARDs, as well as throughout the treatment [10].

Tuberculosis is a disease caused by a *Mycobacterium tuberculosis* infection, which is usually transmitted by air from patients with pulmonary tuberculosis. The latent, asymptomatic infections may persist in some people, who may develop the disease after months or years. The main purpose of diagnosing latent tuberculosis is the possibility of providing medical treatment to prevent tuberculosis. Given that the overall incidence of tuberculosis in Romania is the highest in the European Union, and that patients with autoimmune diseases such as rheumatoid arthritis and ankylosing spondylitis are included in the risk categories, screening for this condition is of major importance.

Rheumatoid arthritis (RA) is a systemic inflammatory condition that affects predominantly the synovial joints. The chronic inflammatory process at this level causes progressive, irreversible joint destruction, with permanent joint deformities, accompanied by a functional deficit. Systemic manifestations such as cardiovascular, pulmonary, renal, neurological, and ocular issues cause a reduction in life expectancy by 5 to 10 years [11,12].

RA is the most common inflammatory rheumatism with a prevalence of about 0.5–1% in the general population, with large variations in some ethnic groups, reaching 5% in some Native American populations. In Romania, there is an estimated number of over 200,000 patients with this condition. RA is 2–3 times more common in females than males, with an annual incidence of 0.5 new cases/1000 inhabitants for women and 0.2 new cases/1000 inhabitants for men. The most common onset of the disease is between 50 and 75 years old [9,13].

The pharmacological treatment of patients with RA is based on the use of DMARDs (disease-modifying antirheumatic drugs), which are remissive medications that relieve symptoms, act in the long term, and that may prevent irreversible structural lesions or slow their progression and the onset of functional deficits. Their classification is the following:

- Synthetic DMARDs (sDMARDs), which are produced by chemical synthesis processes and are divided into the conventional synthetic DMARDs (csDMARDs: Methotrexate, Leflunomide, Sulfasalazine, Hydroxychloroquine, Cyclosporine A, and Azathioprine) and targeted synthetic DMARDs (tsDMARDs: Tofacitinib andBaricitinib).
- Biological DMARDs (bDMARDs) are produced by genetic engineering processes based on living cell cultures and are divided into the originals (boDMARDs) and biosimilars (bsDMARDs), as shown in Table 1 [14].

**Table 1.** Biological DMARDs (bDMARDs) used in RA.

| International Nonproprietary Name | Type of Molecule | Mechanism | Usual Dose in RA | Trade Name |
|---|---|---|---|---|
| ABATACEPT | IgG-Fc fusion protein | T-cell costimulatory signal inhibitor | 125 mg/week | ORENCIA |
| ADALIMUMAB | Human monoclonal antibody | TNF alfa inhibitor | 40 mg/2 weeks s.c. | HUMIRA |
| CERTOLIZUMAB | Fab Fragment of human monoclonal antibody | TNF alfa inhibitor | 200 mg/2 weeks s.c. | CIMZIA |
| ETANERCEPT | IgG-Fc fusion protein | TNF alfa inhibitor | 50 mg/week s.c. | BENEPALI-biosimilar ENBREL-original |
| INFLIXIMAB | Chimeric monoclonal antibody | TNF alfa inhibitor | 3–7.5 mg/kg/4–8 weeks i.v. | REMICADE-biosimilar REMSIMA-original |
| GOLIMUMAB | Human monoclonal antibody | TNF alfa inhibitor | 50 mg/month s.c. | SIMPONI |
| RITUXIMAB | Chimeric monoclonal antibody | Antibody against CD20 on B-cell surface | 1000 mg/6 months i.v. | MABTHERA |
| TOCILIZUMAB | Human monoclonal antibody | IL-6R receptor antagonist | 162 mg/week s.c./i.v. | ROACTEMRA |

Ankylosing spondylitis (AS) is a subset of axial spondyloarthritis and is a chronic inflammatory disease that predominantly affects the spine and peripheral joints. The major feature of the disease is the early damage to the sacroiliac joints. AS evolves to spinal ankylosis. Occasionally, other joints such as the shoulders or hips are involved. Eye and bowel problems may also occur. The main symptom is inflammatory lower back pain, accompanied by stiffness, which improves after exercise [15,16].

The important socioeconomic impact of ankylosing spondylitis is given by the fact that this condition has a prevalence of 0.5–1% and determines a reduction in life expectancy by 5–10 years. All these are associated with the high indirect costs caused by retirement before the age limit, as well as severe disabilities that do not allow for self-care [17].

Medication used to treat ankylosing spondylitis include the following:

A.  Non-steroidal anti-inflammatory drugs (NSAIDs), when used daily at the maximum recommended doses, reduces inflammation, pain, and paravertebral contracture.

B.   Corticosteroids can be used topically for enthesis and peripheral arthritis. Corticosteroid injections into the sacroiliac joints can provide temporary relief of the pain.

C.   Conventional DMARDs (csDMARDs) have no effect on axial manifestations of ankylosing spondylitis, and only limited efficacy in peripheral joints and enthesis. Sulfasalazine is the most widely used drug in the treatment of peripheral ankylosing spondylitis.

D.   Biological therapies (bDMARDs) have shown effectiveness in the reduction of the disease activity as well as stopping the evolution of the disease, allowing the social reintegration of young patients. Biological treatments of ankylosing spondylitis involve the use of TNFα blockers (adalimumab, certolizumab, etanerceptum, golimumab, and infliximab) or interleukin-17A inhibitors (secukinumab) [18].

The TNFα blockers used in AS are Adalimumab (original and biosimilar), Certolizumab pegol, Etanercept (original and biosimilar), and Golimumab and Infliximab (original and biosimilar). They are used in similar doses as for RA.

A special class of bDMARDs for AS are the Interleukin-17A Inhibitors—Secukinumab, which is administered by a subcutaneous injection at a dose of 150 mg/week at weeks 0, 1, 2, and 3, followed by 150 mg every 4 weeks thereafter [10,19].

Prior to initiating biological therapy, patients should be evaluated for the risk of developing a reactivation of latent tuberculosis. Carefully assessing the patient's medical history, performing clinical examinations, lung radiography, and interferon-gamma release assays (IGRA), such as QuantiFERON-TB Gold or the tuberculin skin test (TST), should be performed. Patients with positive TST results (IDR > 5 mm) or positive QuantiFERON results will be recommended chemoprophylaxis with isoniazid at 5 mg/kg/day, with a maximum of 300 mg/day for 9 months. The biological therapy may be initiated after at least one month of chemoprophylaxis. In patients who have had negative initial tests, periodic screening for the reactivation of tuberculosis is recommended at least once every 12 months by using the QuantiFERON-TB Gold test or the tuberculin skin test (TST). The QuantiFERON-TB Gold In-Tube assay seems to be a more accurate test for the detection of LTBI in RA patients compared with the TST [14,20,21].

The QuantiFERON-TB Gold Test (QFT) is a simple blood test that aids in the diagnosis of *Mycobacterium tuberculosis* infections, both for the latent and for the active forms. The test measures the intensity of the cellular immune responses to antigens that mimic ESAT-6 and CFP-10 proteins, produced by the mycobacterium. The recognition process includes the generation and release of cytokines, especially interferon (IFN)-γ, which can be detected and quantified by an enzyme-linked immunosorbent assay [22].

The results of the QuantiFERON-TB Gold test are based on the proportion of interferon-γ released in response to tuberculin as compared with the mitogen. Unlike the tuberculin skin test, QFT results are not influenced by the bacille Calmette-Guerin (BCG) vaccination, and a history of infections with nontuberculous mycobacteria also have a lesser influence. Adverse hypersensitivity reactions that may occur with the tuberculin skin test are also eliminated. Also, the results of the QuantiFERON-TB test are much more objective, without errors in reading and interpretation [22].

The mitogen tube is used as an IFN-γ positive control for each specimen tested and serves as a control for correct blood handling and incubation. The mitogen used is phytohaemagglutinin-P (PHA), which is a nonspecific stimulator of T-cells [22].

A low IFN-γ response to mitogen (<0.5 IU/mL) indicates an indeterminate result when a blood sample also has a negative response to the TB antigens. This pattern may occur with insufficient lymphocytes, reduced lymphocyte activity due to improper specimen handling, or an inability of the patient's lymphocytes to generate IFN-γ [23].

The diagnosis of a latent tuberculosis infection (LTBI) in patients with immune-mediated inflammatory diseases raises a number of issues due to the characteristics of these patients. Although the QFT test has a 99% specificity and an overall sensitivity for active TB of 92%, some patients remain with an "indeterminate result" due to the inability of lymphocytes to secrete interferon (IFN)-γ after 24 h of stimulation to phytohemagglutinin

(PHA), causing the failure of the appropriate positive control. The reported prevalence of indeterminate results is higher in patients with chronic inflammatory conditions compared to healthy people. Also, patients who receive at least one immunosuppressive drug (especially steroids) are more likely to have an indeterminate result [23,24].

Regarding the risk of latent tuberculosis reactivation, numerous studies have concluded that biological therapy is associated with an increased risk of tuberculosis. A latent tuberculosis infection is the body's immune response to contact with *Mycobacterium tuberculosis*. Infected people have no clinical manifestations nor radiological or bacteriological evidence, and they are not contagious, but some of them risk reactivating the disease [25].

A retrospective study conducted in 11 medical centers in Romania on patients with rheumatoid arthritis (RA), ankylosing spondylitis (AS), and psoriatic arthropathy treated with biological anti-TNF alpha therapy between January 1999 and June 2011 investigated the incidence of tuberculosis. The observational research included 693 patients, of whom 492 had RA. The study identified 15 patients diagnosed with tuberculosis, with positive cultures for *Mycobacterium tuberculosis* in 11 cases (73.3%). TB suspicion was histologically confirmed in 40% of cases. The average duration of TB development after the initiation of the TNF-alpha inhibitor treatment was 23.26 months. In 7 out of 12 cases of TB treated with Infliximab, the disease occurred in the first year of treatment [26].

A 2001 report in the New England Journal of Medicine showed that tuberculosis that is associated with biological therapy with Infliximab had a tendency to reactivate after 11 or 12 weeks of treatment, and the disease had an unusual course in 50% of patients, with extrapulmonary manifestations in about 10% of patients with disseminated tuberculosis [27].

A study conducted by J. Harris and J. Keane in 2010 summarizes the reported effects of TNF blockers on the immune responses in patients with RA following the cases of TB reactivation in patients on anti-TNF therapy. Studies on TNF blockers have resulted in a better understanding of the complex interactions between different cells of the immune system in inflammatory and infectious diseases, and it is clear that TNF blockers can interfere with innate and adaptive immunity, causing tuberculosis infections [28].

## 3. Materials and Methods

Our study included 76 patients diagnosed with RA and 63 patients diagnosed with ankylosing spondylitis, both undergoing biological treatments with original and biosimilar bDMARDs, in monotherapy or in combination therapy (with conventional DMARDs and NSAIDs). These patients were monitored for tuberculosis infections by performing the QuantiFERON-TB Gold test.

The data collection was done by consulting the electronic database of the Romanian Register of Rheumatic Diseases for the patients monitored in the Rheumatology Clinic I of the Rehabilitation Hospital in Iași. Based on this information, two databases were prepared, depending on the patient's diagnosis of rheumatoid arthritis or ankylosing spondylitis.

In the rheumatoid arthritis database that included 76 patients who were monitored for tuberculosis between April 2017 and May 2019, the following information was recorded:

- The date of patient evaluation
- The age and gender of the patient
- The biological agent used at the time of the evaluation
- The date of the biological therapy initiation
- The starting date of the biological agent used at the time of the evaluation
- Remissive therapy (conventional DMARDs) and corticosteroid treatment at the time of the evaluation
- Any pathological history of tuberculosis
- QuantiFERON-TB Gold test results.

The database with 63 ankylosing spondylitis patients who were screened for tuberculosis between November 2015 and November 2019 included the following information:

- The date of the patient evaluation

- The age and gender of patient
- The biological agent used at the time of the evaluation
- The date of the biological therapy initiation
- The starting date of biological agent used at the time of the evaluation
- NSAID treatment
- QuantiFERON-TB Gold test results

## 4. Description of the Machine Learning Methods

### 4.1. k-Nearest Neighbor

Instance-based learning reduces the learning effort by simply storing the examples presented to the learning agent and classifying the new instances based on the closeness to their "neighbors", i.e., previously encountered instances (i.e., from the training set) with similar attribute values. The nearest neighbor (NN) algorithm classifies a new instance in the same class as the closest stored instance in the attributed space. A straightforward extension is the *k*-NN, where *k* neighbors (instead of 1) are taken into account when determining the membership of an instance to a class. This approach is especially useful when data are affected by noise and the nearest neighbor of an instance may indicate an erroneous class [29].

The k-nearest neighbor (kNN) algorithm is based on the choice of k-nearest neighbors using a distance function as a criterion. The output is computed by aggregating the outputs of those k training instances. As a distance function, one can use the Euclidean or Manhattan distance, and usually particularizations of the Minkowski distance. Choosing the value of k is important. If k is too small, then the classification can be affected by the noise in the training data, and if the value of k is too large, then distant neighbors can affect the correctness of the results. To avoid the difficulty of finding an optimum value for k, one can weight the neighbor influence. The neighbors have a greater weight as they are closer to the instance, while those farther apart weigh less [30].

The advantage of nearest-neighbor algorithms is the very quick learning (simple storing of instances) and high prediction capability. The disadvantage is the slow process of searching the nearest instance by processing the whole training set.

### 4.2. Non-Nested Generalized Exemplar (NNGE)

The nested generalized exemplar (NGE) theory [31] is an incremental form of inductive learning from examples by extending the nearest neighbor classification method. NGE is a learning paradigm based on class exemplars, where an induced hypothesis has the graphical shape of a set of hyper-rectangles in an *n*-dimensional Euclidean space. Exemplars of classes are either hyper-rectangles or single training instances, i.e., points. Wettschereck and Dietterich [32] showed that the performance of NGE is poor on many problems, mainly because of overgeneralization. It was hypothesized that these issues were caused by allowing hyper-rectangles to nest or overlap. Martin [33] proposed a solution to avoid all forms of overgeneralization by never allowing exemplars to nest or overlap in an algorithm called the non-nested generalized exemplar (NNGE). Nesting or overlapping is prevented by testing each prospective new generalization to ensure that it does not cover any negative examples, and by modifying any generalizations that are later found to do so. It always tries to generalize new examples to their nearest neighbor of the same class, but if this is impossible due to intervening negative examples, no generalization is performed. If a generalization later conflicts with a negative example, the generalization is modified to maintain consistency. In the variant implemented in WEKA [34], NNGE uses the IB4 [35] attribute weighting scheme and mutual information [36] and gives up the weighting of exemplars.

### 4.3. C4.5 Algorithm

In the case of the simple inductive learning structure of a decision tree, the general rule of membership of an instance to a class is given by traversing the tree branches from

the root to the leaf corresponding to the instance or class. Each internal node of the tree represents a test of none or more properties, and the branches that go from that node are labeled with the possible results of the test. C4.5 is such a decision tree induction algorithm, and it recursively partitions the data set and selects the test which leads to the highest information gain [37], i.e., the resulting sub-classes should be as homogenous as possible. In order to avoid over-fitting, the resulting tree can be pruned at the end of the categorization process. In this way, the tree will be smaller, with more errors on the training set than the unpruned version, but it supposedly will have better generalization capabilities.

The C4.5 algorithm generates a decision tree for the given dataset by the recursive partitioning of data [37]. The algorithm supports tree pruning at the end of the training process. In our experiments, we tested both the pruned and unpruned variants. Even if the pruned decision trees were in some cases smaller by 3–4 nodes or leaves, the differences compared to the unpruned versions were not significant. Usually, in the unpruned version, the accuracy on the training sets are 1–2% higher, and the accuracy on the testing sets are 1–2% lower than in the pruned version.

### 4.4. Random Forest

The random tree (RT) classifier builds a tree that considers k random features at each node and performs no pruning. Therefore, its error rate on the training set alone is rather small. A random forest (RF) [38] is composed of several random trees. Each such tree is built on a slightly different variant of the training set. Such training sets are created using bagging, where the same number of instances as the number in the original training set are randomly selected with a replacement. The trees are constructed using only a random subset of attributes when creating the node partitions. The partitioning continues until all instances are classified by the leaves. To classify a new instance, the input vector is run down each of the trees in the forest. Each tree gives a classification, and this is considered to be a "vote" for that class. The forest chooses the classification with most of the votes, over all the trees in the forest.

Random forest can be used for both regression and classification problems. Cross-validation is no longer necessary, because RF gives an internal estimate of the generalization error. The available variable importance measures can be used for variable selection. RF produces proximities that can provide a wealth of information through novel visualizations of the data. Proximities can also be used to impute missing values. RF can be used successfully for a wide variety of applications in several disciplines [39].

### 4.5. Support Vector Machines (SVM)

Support vector machines [40] represent a method of classification (binary classification in the standard approach) and regression. An SVM model considers the training instances as points in a multi-dimensional space, which can be transformed in order for the classes to be separated with a large margin. The idea of splitting the hyperspace into two parts can be also found in the training principle of the single-layer perceptron, for example, but in this case, it works only if the problem is linearly separable.

For the nonlinear cases, SVM uses kernels for mapping the data into a different space with more dimensions compared with the original space, where a problem can become linearly separable even if it was not originally so. In addition, some errors in the classification of the training data can be allowed using soft margins with the goal of increasing the generalization capability.

Support vector machines benefit from solid mathematical foundations, which offer a very good accuracy compared with other learning methods. Another advantage is the small number of parameters that the user has to choose (the type of kernel with its parameters, and a cost parameter that defines the balance between the tolerance for training errors and the generalization capability). A small disadvantage is the fact that the standard model is binary, and in order to apply it to problems with multiple classes, it is necessary to obtain

several partial models that are subsequently aggregated on the basis of various strategies such as "one-versus-all" or "one-versus-one".

The process of learning in traditional classifiers is based on the empirical risk minimization (ERM) principle. That is, the model that obtains the lowest error on the training set is chosen. One problem with this principle is that the noise in the training data is ignored; thus, in real-world problems, the ERM-based models such as the traditional artificial neural networks will learn not only the data but also the noise, leading to inaccurate predictions. Another common issue with conventional learners is that there are many parameters to be tuned. The third problem that usually occurs is that the algorithms converge only to local optima within their search space, while the global optimum is desired.

The SVM method proposed by Vapnik has several advantages over the other learning methods. SVM is based on the structural risk minimization principle from the computational learning theory which always converges to a global optimum, in contrast with ERM. Additionally, SVM has strong generalization capabilities. As a disadvantage, SVM models are computationally expensive; they need time and memory as the complexity of the model increases (depending on the dimension of the training data) [41].

Consequently, SVM, as statistical learning theory-based machine learning method, gained recognition due to its features and its promising generalization performance, such as (i) the ability to model non-linear relationships; (ii) the dimensionality of the input space does not affect the generalization; and (iii) the loss function is related to a quadratic programming problem whose solution is global and, in general, unique [42].

### 4.6. ReliefF

ReliefF [43] is a well-known non-parametric feature-weighting algorithm. It is a local algorithm computes the relevance of a feature at a sample in terms of the difference between that sample and the other nearby samples of the same class and the other nearby samples of other classes. These differences are related to the ability of an attribute to perform a classification by itself by linearly separating the classes. If more instances are grouped together on an attribute axis and separated from the instances that belong to the other class, then that particular attribute is more important for the classification task. The relevance of an attribute overall is the average relevance of the attribute across all training samples [44].

ReliefF is more robust than the original Relief algorithm. For every target sample, it selects the nearby hits and nearby misses and averages their distances. The relevance to the features in both the Relief and ReliefF algorithms is assigned based on the ability to disambiguate similar samples. In this case, similarity is defined by proximity in feature space. The relevant features will accumulate high positive weights. The irrelevant features will have near-zero weights [44].

### 4.7. Information Gain

The information gain criterion for feature selection is based on ideas about the homogeneity of the instances with a certain attribute value that belong to a certain class. It is also used in decision tree algorithms such as ID3 [45] and C4.5 [37]. The main idea is that the importance of an attribute is related to its capability to solve the classification problem by itself, i.e., to what extent it is possible to reach a complete classification by testing only the values of the considered attribute. This is related to the use of entropy as a homogeneity measure. For an attribute value, the maximum value of the entropy means that the instances are equally distributed among classes (which is not helpful for classification), while a value of 0 for the entropy means that all instances with that attribute value belong to the same class (which helps the classification). For an attribute, a weighted average of the entropy corresponding to the individual attribute values can be computed. The information gain for an attribute is defined as the decrease in entropy by splitting the dataset according to that attribute. The higher the information gain, the more important an attribute is.

*4.8. Correlation-Based Feature Selection*

The correlation-based feature selection (CFS) method [46] is based on the idea that good feature sets contain features that are highly correlated with the class, but not correlated with one another. It addresses the problem of feature selection through a correlation-based approach. The central hypothesis is that good feature sets contain features that are highly correlated with the class, yet uncorrelated with each other. A feature evaluation formula inspired from test theory is employed. CFS couples this evaluation formula with an appropriate correlation measure and a heuristic search strategy, e.g., a greedy stepwise forward search.

## 5. Experimental Study

*5.1. Medical Data Analyses*

For the database with rheumatoid arthritis patients, the following data were taken into account:

- Input variables: the date of the patient evaluation, the age and gender of patient, the biological agent used at the time of the evaluation, the date of the biological therapy initiation, the starting date of biological agent used at the time of the evaluation, remissive therapy (conventional DMARDs), corticosteroid therapy, a pathological history of tuberculosis.
- Output variable: QuantiFERON-TB Gold test result.

For the database with ankylosing spondylitis patients, the following were taken into account:

- Input variables: the date of the patient evaluation, the age and gender of patient, the biological agent used at the time of the evaluation, the date of the biological therapy initiation, the starting date of biological agent used at the time of the evaluation, NSAID treatment.
- Output variable: QuantiFERON-TB Gold test result.

All patients with an initially negative QuantiFERON test result and who tested positive during the biological treatment were subsequently diagnosed with tuberculosis in the Pneumology Clinic and they received specific treatment. The reason why we considered the positive QuantiFERON test result as evidence of a reactivation of latent tuberculosis was that it was later confirmed by a specialist, and it was the first test to indicate this diagnosis, as it was performed as a screening method in patients undergoing biological treatment. Because we wanted to analyze whether the duration of the biological treatment influences the reactivation of TB, we decided to take into account the date of the positive QuantiFERON result and not the later date when the diagnosis was confirmed by the pneumologist.

### 5.1.1. Rheumatoid Arthritis Database Results

Out of the total of 76 patients with rheumatoid arthritis, 80.3% (n = 61) were women and 19.7% (n = 15) were men. The age of the patients ranged between 19 and 81 years old.

At the time of inclusion in the study, the distribution of RA patients by gender and the biological therapy is given in Table 2.

**Table 2.** Distribution of RA patients by gender and the biological therapy followed at the time of inclusion in the study.

| Trade Name | International Nonproprietary Name | Patients (%) | Women (nr., %) | Men (nr., %) |
|---|---|---|---|---|
| BENEPALI | Etanercept (biosimilar) | n = 3 (3.95) | n = 3 (3.95) | n = 0 (0) |
| CIMZIA | Certolizumab | n = 3 (3.95) | n = 3 (3.95) | n = 0 (0) |
| ENBREL | Etanercept (original) | n = 13 (17.11) | n = 8 (10.53) | n = 5 (6.58) |
| HUMIRA | Adalimumab | n = 14 (18.42) | n = 12 (15.79) | n = 2 (2.63) |
| MABTHERA | Rituximab | n = 20 (26.32) | n = 18 (23.68) | n = 2 (2.63) |
| ORENCIA | Abatacept | n = 2 (2.63) | n = 2 (2.67) | n = 0 (0) |
| REMICADE | Infliximab (biosimilar) | n = 1 (1.31) | n = 0 (0) | n = 1 (1.31) |
| REMSIMA | Infliximab (original) | n = 1 (1.31) | n = 1 (1.31) | n = 0 (0) |
| ROACTEMRA | Tocilizumab | n = 16 (21.05) | n = 12 (15.79) | n = 4 (5.26) |
| SIMPONI | Golimumab | n = 3 (3.95) | n = 2 (2.63) | n = 1 (1.31) |
| Total | | n = 76 (100) | n = 61 (80.3) | n = 15 (19.7) |

n = number of cases expressed as absolute value (percentage of the total number of cases).

During the study period, 7 patients (9.21%), 5 women and 2 men, switched to another biological treatment, as follows:

- A patient switched from Simponi (Golimumab) to Roactemra (Tocilizumab);
- A patient switched from Mabthera (Rituximab) to Roactemra (Tocilizumab);
- A patient switched from Orencia (Abatacept) to Benepali (biosimilar Etanercept);
- A patient switched from Cimzia (Certolizumab) to Roactemra (Tocilizumab);
- A patient switched from Humira (Adalimumab) to Roactemra (Tocilizumab);
- A patient switched from Humira (Adalimumab) to Enbrel (Etanercept);
- A patient switched from Enbrel (Etanercept) to Humira (Adalimumab).

At the end of the study, the distribution of patients by gender and the biological therapy is presented in Table 3.

**Table 3.** Distribution of RA patients by gender and the biological therapy followed at the end of the study.

| Trade Name | International Nonproprietary Name | Patients (%) | Women (nr., %) | Men (nr., %) |
|---|---|---|---|---|
| BENEPALI | Etanercept (biosimilar) | n = 4 (5.26) | n = 4 (5.26) | n = 0 (0) |
| CIMZIA | Certolizumab | n = 2 (2.63) | n = 2 (2.63) | n = 0 (0) |
| ENBREL | Etanercept (original) | n = 13 (17.11) | n = 8 (10.53) | n = 5 (6.58) |
| HUMIRA | Adalimumab | n = 13 (17.11) | n = 11 (14.47) | n = 2 (2.63) |
| MABTHERA | Rituximab | n = 19 (25) | n = 17 (22.37) | n = 2 (2.63) |
| ORENCIA | Abatacept | n = 1 (1.31) | n = 1 (1.31) | n = 0 (0) |
| REMICADE | Infliximab (biosimilar) | n = 1 (1.31) | n = 0 (0) | n = 1 (1.31) |
| REMSIMA | Infliximab (original) | n = 1 (1.31) | n = 1 (1.31) | n = 0 (0) |
| ROACTEMRA | Tocilizumab | n = 20 (26.32) | n = 16 (21.05) | n = 4 (5.26) |
| SIMPONI | Golimumab | n = 2 (2.63) | n = 1 (1.31) | n = 1 (1.31) |
| Total | | n = 76 (100) | n = 61 (80) | n = 15 (20) |

n = number of cases expressed as absolute value (percentage of the total number of cases).

Out of a total of 76 patients, 9.21% (n = 7) had a history of tuberculosis infection and 3.95% (n = 3) had a positive QuantiFERON-TB Gold test result prior to the beginning of biological therapy.

Among the RA patients with negative QFT test results at the initial evaluation, 21.05% (n = 16) of them had positive results at QuantiFERON-TB Gold testing during the biological treatment that suggested the reactivation of latent tuberculosis. The diagnosis was confirmed in the pneumology clinic, and all patients received specific treatment. Table 4 contains the distribution of RA patients with a positive QFT test result depending on the biological therapy.

**Table 4.** Distribution of RA patients with positive QFT test result depending on the biological therapy.

| Trade Name | International Nonproprietary Name | Patients (%) |
|---|---|---|
| MABTHERA | Rituximab | n = 5 (31.25) |
| ROACTEMRA | Tocilizumab | n = 4 (25) |
| ENBREL | Etanercept (original) | n = 4 (25) |
| HUMIRA | Adalimumab | n = 2 (12.5) |
| CIMZIA | Certolizumab | n = 1 (6.25) |
| Total | | n = 16 (100) |

n = number of cases expressed as absolute value (percentage of total number of cases).

### 5.1.2. Ankylosing Spondilytis Database Results

Out of a total of 63 patients with ankylosing spondylitis, 23.8% (n = 15) were women and 76.2% (n = 48) were men. The age of the patients ranged between 23 and 79 years old.

At the time of inclusion in the study, the distribution of the AS patients by gender and the biological therapy is given in Table 5.

**Table 5.** Distribution of AS patients by gender and the biological therapy followed at the time of inclusion in the study.

| Trade Name | International Nonproprietary Name | Patients (%) | Women (nr., %) | Men (nr., %) |
|---|---|---|---|---|
| BENEPALI | Etanercept (biosimilar) | n = 3 (4.76) | n = 0 (0) | n = 3 (4.76) |
| COSENTYX | Secukinumab | n = 3 (4.76) | n = 1 (1.59) | n = 2 (3.18) |
| ENBREL | Etanercept (original) | n = 17 (26.99) | n = 4 (6.35) | n = 13 (20.64) |
| HUMIRA | Adalimumab | n = 22 (34.92) | n = 5 (7.94) | n = 17 (26.98) |
| REMICADE | Infliximab (biosimilar) | n = 11 (17.46) | n = 1 (1.59) | n = 10 (15.87) |
| SIMPONI | Golimumab | n = 7 (11.11) | n = 4 (6.35) | n = 3 (4.76) |
| Total | | n = 63 (100) | n = 15 (23.8) | n = 48 (76.2) |

n = number of cases expressed as absolute value (percentage of total number of cases).

Eighty-seven point three percent (n = 55) of patients received biological monotherapy, while 12.7% (n = 8) patients received NSAIDs with bDMARDs.

Twenty-three point eight percent (n = 15) of AS patients, all undergoing monotherapy, had positive QFT test results, while 76.2% (48) had negative results at the time of the evaluation. The reactivation of TB was confirmed in the pneumology clinic for all 15 patients and specific treatment was prescribed.

The distribution of AS patients with positive QFT test results depended on the biological therapy they followed at the time of the evaluation (Table 6).

**Table 6.** Distribution of AS patients with positive QFT test results depending on the biological therapy.

| Trade Name | International Nonproprietary Name | Patients (%) |
|---|---|---|
| BENEPALI | Etanercept (biosimilar) | n = 1 (6.67) |
| ENBREL | Etanercept (original) | n = 6 (40) |
| HUMIRA | Adalimumab | n = 2 (13.33) |
| REMICADE | Infliximab (biosimilar) | n = 5 (33.33) |
| SIMPONI | Golimumab | n = 1 (6.67) |
| Total | | n = 15 (100) |

n = number of cases expressed as absolute value (percentage of total number of cases).

## 5.2. Modelling Results

### 5.2.1. Rheumatoid Arthritis Results

The aim of our study regarding the patients with rheumatoid arthritis was to identify the reactivation of latent tuberculosis, evidenced by the positive QuantiFERON test result in patients undergoing biological therapy. We aimed to establish the factors that influenced the reactivation of tuberculosis, taking into account both elements related to the patient (age, gender, and medical history) and the treatment they followed (the type of biological therapy, the use of conventional DMARDs or corticosteroids, and the duration of therapy) in order to determine which of them were relevant for our problem.

The problem had the following input attributes: age (integer value, in years), gender (binary value), biological therapy (symbolic, with the following types: 1 = BENEPALI, 2 = CIMZIA, 3 = ENBREL, 4 = HUMIRA, 5 = MABTHERA, 6 = ORENCIA, 7 = REMICADE, 8 = REMSIMA, 9 = ROACTEMRA, 10 = SIMPONI), the difference from the beginning of therapy (integer value, in days), the use of remissive therapy (conventional DMARDs) or corticosteroid therapy (AZA, CYCLOSPORINE, HCQ, LEF, MTX, SSZ, PDN < 7.5, all with binary values) and the presence of a history of tuberculosis (binary value). The output was also binary and consisted of the QuantiFERON-TB Gold Test indicator.

One of the difficult elements of the analysis was that the dataset with RA patients contained multiple records for some patients and single records for others, and the number of multiple records was not constant. For the patients with multiple records, the starting date of the biological therapy was unique, while the dates of the medical tests differed for each record.

Therefore, several approaches to classification were proposed, where an important process is data preprocessing:

- The transformation of the calendar dates to a format suitable for the application of the classification algorithms, which involved transforming them into a difference of days between the date of the medical tests and the starting date of the biological therapy
- Considering all records as independent, and the direct application of the classification algorithms
- Considering only the latest, most recent record for each patient, and the application of the classification algorithms
- Considering the dynamic evolution of patients, i.e., the creation of a distinct model for the patients who tested positive during treatment, i.e., for whom the QuantiFERON indicator was initially 0 and, at one point, became 1, suggesting the presence of a tuberculosis infection
- The application of the attribute selection algorithms, which would identify a subset of more relevant entries, and the application of classification algorithms on this smaller set of indicators.

For classification, the classical algorithms used generally gave good results with the implementation in WEKA [47], a popular collection of machine learning algorithms: random forest, nearest neighbor (NN), k-nearest neighbors (kNN), C4.5 decision trees,

and support vector machines (SVM). For each algorithm, the most important parameters varied as follows: the number of trees for random forest, the number of neighbors and the weighting function of the importance of neighbors for k-nearest neighbors, pruned or unpruned versions for C4.5 decision trees, and the kernel type for support vector machines.

The results obtained are presented in three variants: on the entire training set, to highlight the ability of a model to learn the data, and with a 10-fold cross-validation and the leave-one-out approach, to highlight the generalization capability of the learned model.

The first analysis directly considered all the records in the training set. This analysis was the baseline for comparing results. Given that some patients have only one record and others have a larger, variable number of records, the training instances were not all independent. Table 7 presents the results obtained with this approach. The text in bold represents the best result.

**Table 7.** Classification results for all records in the dataset.

| Classification Method | Accuracy on the Training Set | Accuracy for Cross-Validation | Accuracy for Leave-One-Out |
|---|---|---|---|
| Random forest, 100 trees | 100% | 80% | 80% |
| Random forest, 1000 trees | 100% | 81.7391% | 82.6087% |
| NN | 100% | 80% | 79.1304% |
| **kNN, k = 20, w = 1/d** | **100%** | **84.3478%** | **83.4783%** |
| kNN, k = 100, w = 1/d | 100% | 81.7391% | 76.5217% |
| C4.5, Pruned | 80% | 80% | 80% |
| C4.5, Unpruned | 92.1739% | 77.3913% | 74.7826% |
| SVM, Puk kernel | 85.2174% | 78.2609% | 78.2609% |

Next, only the latest, most recent records for each patient were considered. In this way, it is ensured that the training data were independent because each patient had an equal weight in the development of the model. Table 8 presents the results obtained in this case, with kNN method proving the best result. One can see that the results are worse than those in Table 7, especially for the cross-validation procedure, because redundant patient information is removed.

Therefore, the dataset was augmented to increase the number of instances in class 1. The new instances resulted from slightly perturbing the values for age and the difference between the beginning and end of therapy, while keeping the value for the class the same. Table 10 presents the results of the precision and recall analysis with the augmented dataset. One can notice much improvement compared to the case of Table 9, especially the random forest model, which succeeds in finding a good balance between the two classes and the final F1 score, combining precision and recall to be the best model compared to the other studied models.

**Table 8.** Classification results when only the latest record for each patient is retained.

| Classification Method | Accuracy on the Training Set | Accuracy for Cross-Validation | Accuracy for Leave-One-Out |
|---|---|---|---|
| Random forest, 100 trees | 100% | 57.8313% | 62.6506% |
| Random forest, 1000 trees | 100% | 65.0602% | 65.0602% |
| NN | 100% | 63.8554% | 67.4699% |
| kNN, k = 20, w = 1/d | 100% | 72.2892% | 72.2892% |
| **kNN, k = 83 (all), w = 1/d** | **100%** | **84.6988%** | **74.6988%** |
| C4.5, Pruned | 74.6988% | 69.8795% | 72.2892% |
| C4.5, Unpruned | 85.5422% | 61.4458% | 67.4699% |
| SVM, Puk kernel | 89.1566% | 69.8795% | 67.4699% |

For this dataset, additional experiments were performed to assess the performance of the algorithms from the point of view of precision and recall. Since the dataset was unbalanced (62 instances in class 0 and 21 instances in class 1), such analyses can provide additional information beyond the assessment of accuracy. For the best representative of each class of machine learning methods, the following indicators were computed: the true positive (TP) rate, the false positive (FP) rate, precision, recall, and the F1 score. In each cell, the first value is for class 0, the second one is for class 1, and the third one is the weighted average of the first two values.

In Table 9, one can see that all methods fail to capture the patterns related to class 1, except for random forest, which manages to handle class 1 to a small extent.

**Table 9.** Precision and recall analysis for the original dataset.

| Classification Method | TP Rate | FP Rate | Precision | Recall | F1 |
|---|---|---|---|---|---|
| Random forest, 1000 trees | 0.839<br>0.095<br>0.651 | 0.905<br>0.161<br>0.717 | 0.732<br>0.167<br>0.589 | 0.839<br>0.095<br>0.651 | 0.782<br>0.121<br>0.615 |
| kNN, k = 83 (all), w = 1/d | 1<br>0<br>0.747 | 1<br>0<br>0.747 | 0.747<br>N/A<br>N/A | 1<br>0<br>0.747 | 0.855<br>N/A<br>N/A |
| C4.5, Pruned | 0.935<br>0<br>0.699 | 1.000<br>0.065<br>0.763 | 0.734<br>0<br>0.548 | 0.935<br>0<br>0.699 | 0.823<br>0<br>0.615 |
| SVM, Puk kernel | 0.935<br>0<br>0.699 | 1<br>0.065<br>0.763 | 0.734<br>0<br>0.548 | 0.935<br>0<br>0.699 | 0.823<br>0<br>0.615 |

Another analysis referred to the evolution of the positive patients, for which the QuantiFERON indicator was initially 0 and, at one point, became 1. For these patients, a new class was included in the model, called "P". In this case, we wanted to highlight a symbolic model for this medical situation. Beyond the numerical results, the aim was to provide medical staff with some explanations for the conditions that could change the health of patients in the sense mentioned above.

Table 11 shows the results obtained with the NNGE (non-nested generalized exemplars) algorithm, which is based on the instance-based learning paradigm, but determines generalized specimens, i.e., hyper-rectangles in the problem space, consisting of a set of individual instances that belong to the same class and cannot contain instances of a different class. These hyper-rectangles can be interpreted as rules.

**Table 10.** Precision and recall analysis for the augmented dataset.

| Classification Method | TP Rate | FP Rate | Precision | Recall | F1 |
|---|---|---|---|---|---|
| Random forest, 1000 trees | 0.645 | 0 | 1 | 0.645 | 0.784 |
| | 1 | 0.355 | 0.741 | 1 | 0.851 |
| | 0.824 | 0.179 | 0.87 | 0.824 | 0.818 |
| kNN, k = 124 (all), w = 1/d | 0.435 | 0 | 1 | 0.435 | 0.607 |
| | 1 | 0.565 | 0.643 | 1 | 0.783 |
| | 0.72 | 0.285 | 0.82 | 0.72 | 0.695 |
| C4.5, Pruned | 0.532 | 0.127 | 0.805 | 0.532 | 0.641 |
| | 0.873 | 0.468 | 0.655 | 0.873 | 0.748 |
| | 0.704 | 0.299 | 0.729 | 0.704 | 0.695 |
| SVM, Puk kernel | 0.694 | 0.127 | 0.843 | 0.694 | 0.761 |
| | 0.873 | 0.306 | 0.743 | 0.873 | 1 |
| | 0.784 | 0 | 1 | 0.784 | 0.782 |

**Table 11.** Model of rules for patient positivation, obtained with the NNGE algorithm.

IF: age = 51.0 ˆ gender in {1} ˆ bio_therapy in {9} ˆ diff_beginning = 531.0 ˆ AZA in {0} ˆ CYCLOSPORINE in {0} ˆ HCQ in {0} ˆ LEF in {0} ˆ MTX in {1} ˆ SSZ in {0} ˆ PDN75 in {0} ˆ tb_history in {0} (1)

IF: 52.0 <= age <= 60.0 ˆ gender in {0} ˆ bio_therapy in {4,5} ˆ 1948.0 <= diff_beginning <= 2653.0 ˆ AZA in {0} ˆ CYCLOSPORINE in {0} ˆ HCQ in {0} ˆ LEF in {0} ˆ MTX in {1} ˆ SSZ in {0} ˆ PDN75 in {0} ˆ tb_history in {0} (3)

Moreover, the C4.5 algorithm was applied again to determine a decision tree (Table 12).

**Table 12.** Decision tree for the positivation of QuantiFERON-TB Gold Test, obtained with the C4.5 algorithm.

```
MTX = 0
|  HCQ = 0
|  |  bio_therapy = 1: 0 (4.0)
|  |  bio_therapy = 2: 0 (3.0/1.0)
|  |  bio_therapy = 3
|  |  |  LEF = 0: 1 (2.0)
|  |  |  LEF = 1: 0 (6.0/2.0)
|  |  bio_therapy = 4: 0 (9.0)
|  |  bio_therapy = 5: 0 (11.0/4.0)
|  |  bio_therapy = 6: 0 (2.0)
|  |  bio_therapy = 7: 0 (0.0)
|  |  bio_therapy = 8: 0 (1.0)
|  |  bio_therapy = 9
|  |  |  age <= 72
|  |  |  |  age <= 63: 0 (4.0/1.0)
|  |  |  |  age > 63: 1 (4.0)
|  |  |  age > 72: 0 (4.0)
|  |  bio_therapy = 10: 0 (1.0)
|  HCQ = 1: 0 (12.0)
MTX = 1
|  bio_therapy = 1: 0 (0.0)
|  bio_therapy = 2: 0 (0.0)
|  bio_therapy = 3: 0 (3.0/1.0)
|  bio_therapy = 4
|  |  age <= 51: 0 (2.0)
|  |  age > 51: P (3.0/1.0)
|  bio_therapy = 5: 0 (6.0/2.0)
|  bio_therapy = 6: 0 (0.0)
|  bio_therapy = 7: 0 (1.0)
|  bio_therapy = 8: 0 (0.0)
|  bio_therapy = 9: 0 (4.0/1.0)
|  bio_therapy = 10: 0 (1.0)
```

These tables present explicit decision rules or trees. While some recent methods, e.g., deep neural networks, provide black box models which are hard to explain, these models can be inspected by practitioners and thus their suggestions for medical decisions can be verified.

The same problem of determining an explicit model for QuantiFERON-TB Gold Test positivation was also handled by first selecting a subset of relevant attributes, proposed by the medical expert: biotherapy, age, gender, and tb_history. The obtained results are presented in Tables 13 and 14.

**Table 13.** Model of rules for the positivation of QuantiFERON-TB Gold Test with attribute selection, obtained with the NNGE algorithm.

```
class P IF: 51.0 <= age <= 52.0 ˆ gender in {0,1} ˆ bio_therapy in {4,9} ˆ tb_history in {0} (2)
class P IF: age = 60.0 ˆ gender in {0} ˆ bio_therapy in {5} ˆ tb_history in {0} (1)
class P IF: age = 55.0 ˆ gender in {0} ˆ bio_therapy in {4} ˆ tb_history in {0} (1)

class 0 IF: 33.0 <= age <= 46.0 ˆ gender in {0,1} ˆ bio_therapy in {9,10} ˆ tb_history in {0} (3)
class 0 IF: 58.0 <= age <= 61.0 ˆ gender in {1} ˆ bio_therapy in {4,7,9,10} ˆ tb_history in {0} (4)
class 0 IF: 58.0 <= age <= 59.0 ˆ gender in {0} ˆ bio_therapy in {1,3,9,10} ˆ tb_history in {0} (6)
class 0 IF: 73.0 <= age <= 81.0 ˆ gender in {0} ˆ bio_therapy in {9} ˆ tb_history in {0} (4)
class 0 IF: age = 65.0 ˆ gender in {1} ˆ bio_therapy in {9} ˆ tb_history in {0} (1)
class 0 IF: age = 62.0 ˆ gender in {0} ˆ bio_therapy in {9} ˆ tb_history in {0} (1)
class 0 IF: age = 67.0 ˆ gender in {0} ˆ bio_therapy in {9} ˆ tb_history in {0} (1)
class 0 IF: 58.0 <= age <= 64.0 ˆ gender in {0} ˆ bio_therapy in {9} ˆ tb_history in {1} (2)
class 0 IF: age = 69.0 ˆ gender in {0} ˆ bio_therapy in {5} ˆ tb_history in {1} (1)
class 0 IF: 62.0 <= age <= 74.0 ˆ gender in {0,1} ˆ bio_therapy in {4,6} ˆ tb_history in {0} (8)
class 0 IF: age = 45.0 ˆ gender in {0} ˆ bio_therapy in {5,8} ˆ tb_history in {0} (2)
class 0 IF: 70.0 <= age <= 80.0 ˆ gender in {0} ˆ bio_therapy in {1,3,5} ˆ tb_history in {0} (5)
class 0 IF: 49.0 <= age <= 56.0 ˆ gender in {0,1} ˆ bio_therapy in {2,3,5} ˆ tb_history in {0,1} (9)
class 0 IF: age = 66.0 ˆ gender in {0} ˆ bio_therapy in {5} ˆ tb_history in {0} (1)
class 0 IF: age = 68.0 ˆ gender in {1} ˆ bio_therapy in {5} ˆ tb_history in {0} (1)
class 0 IF: 61.0 <= age <= 63.0 ˆ gender in {0} ˆ bio_therapy in {1,5} ˆ tb_history in {0} (3)
class 0 IF: 20.0 <= age <= 41.0 ˆ gender in {0,1} ˆ bio_therapy in {3,4,5} ˆ tb_history in {0} (6)
class 0 IF: 49.0 <= age <= 50.0 ˆ gender in {0} ˆ bio_therapy in {4} ˆ tb_history in {0} (3)
class 0 IF: age = 53.0 ˆ gender in {0} ˆ bio_therapy in {4} ˆ tb_history in {0} (1)

class 1 IF: age = 19.0 ˆ gender in {1} ˆ bio_therapy in {3} ˆ tb_history in {0} (1)
class 1 IF: 64.0 <= age <= 65.0 ˆ gender in {0} ˆ bio_therapy in {2,5,9} ˆ tb_history in {0} (4)
class 1 IF: age = 67.0 ˆ gender in {0} ˆ bio_therapy in {9} ˆ tb_history in {1} (1)
class 1 IF: age = 47.0 ˆ gender in {0} ˆ bio_therapy in {5} ˆ tb_history in {1} (1)
class 1 IF: 68.0 <= age <= 69.0 ˆ gender in {0} ˆ bio_therapy in {3,5} ˆ tb_history in {0} (3)
class 1 IF: age = 60.0 ˆ gender in {0} ˆ bio_therapy in {9} ˆ tb_history in {0} (1)
class 1 IF: age = 72.0 ˆ gender in {0} ˆ bio_therapy in {9} ˆ tb_history in {0} (1)
class 1 IF: age = 60.0 ˆ gender in {1} ˆ bio_therapy in {3} ˆ tb_history in {0} (1)
class 1 IF: age = 78.0 ˆ gender in {0} ˆ bio_therapy in {4} ˆ tb_history in {0} (1)
class 1 IF: 43.0 <= age <= 45.0 ˆ gender in {0} ˆ bio_therapy in {3} ˆ tb_history in {0,1} (2)
class 1 IF: age = 57.0 ˆ gender in {0} ˆ bio_therapy in {5} ˆ tb_history in {0} (1)
```

The final analysis was based on the application of attribute selection algorithms to confirm the assumptions resulting from the experience of the medical expert with the most relevant features that were automatically determined for the classification problem. These algorithms were not used for the standard dimensionality reduction since the number of attributes was already manageable. In Table 15, one can see the results provided by three specialized algorithms, which indicate that the attributes considered above are relevant to our problem.

**Table 14.** Decision tree for the positivation of QuantiFERON-TB Gold Test with attribute selection, obtained with the C4.5 algorithm.

```
bio_therapy = 1: 0 (4.0)
bio_therapy = 2: 0 (3.0/1.0)
bio_therapy = 3: 0 (14.0/5.0)
bio_therapy = 4
| gender = 0
| | age <= 58
| | | age <= 51: 0 (3.0)
| | | age > 51: P (3.0/1.0)
| | age > 58: 0 (6.0/1.0)
| gender = 1: 0 (3.0)
bio_therapy = 5
| gender = 0
| | tb_history = 0
| | | age <= 56: 0 (6.0)
| | | age > 56
| | | | age <= 60: 1 (2.0/1.0)
| | | | age > 60: 0 (8.0/3.0)
| | tb_history = 1: 0 (2.0/1.0)
| gender = 1: 0 (2.0)
bio_therapy = 6: 0 (2.0)
bio_therapy = 7: 0 (1.0)
bio_therapy = 8: 0 (1.0)
bio_therapy = 9: 0 (20.0/6.0)
bio_therapy = 10: 0 (3.0)
```

**Table 15.** The results of the attribute selection algorithms.

| InfoGain | ReliefF | CFS |
|---|---|---|
| 0.1196 bio_therapy | 0.1722 LEF | bio_therapy |
| 0.10552 MTX | 0.16894 MTX | HCQ |
| 0.06661 HCQ | 0.10177 HCQ | MTX |
| 0.04665 LEF | 0.04593 gender | SSZ |
| 0.03713 SSZ | 0.03653 bio_therapy | |
| 0.02178 tb_history | 0.02662 SSZ | |
| 0.00968 gender | 0.00907 diff_beginning | |
| 0.00511 CYCLOSPORINE | 0.00856 age | |
| 0.00511 AZA | 0.00808 tb_history | |
| 0.00464 PDN75 | −0.0012 AZA | |
| 0 diff_beginning | −0.0012 CYCLOSPORINE | |
| 0 age | −0.00711 PDN75 | |

Two of the attribute selection algorithms (InfoGain and CFS) concluded that the biological therapy (bDMARDs) and some of the csDMARDs (Methotrexate, Hydroxychloroquine, Sulfasalazine, and Leflunomide), played an important role in developing latent TB in the RA patients treated with the biological therapy. These results are consistent with information from medical studies, as it is known that biological therapy involves an increased risk of the reactivation of latent TB in patients with rheumatoid arthritis. The risk of a mycobacterial infection is also increased in RA patients treated with csDMARDs, but the risk is lower than in those receiving bDMARDs.

The third attribute selection algorithm (ReliefF) concluded that three of the most frequently used csDMARDs (Leflunomide, Methotrexate, and Hydroxychloroquine) played the most important role in developing LTB. This algorithm also considered that the gender of the patient is important.

5.2.2. Ankylosing Spondylitis Results

For the ankylosing spondylitis database, the aim of the modelling was also to identify the reactivation of latent tuberculosis in patients undergoing biological therapy, by considering the positive result of the QuantiFERON test. The relevant factors for our problem were selected from elements regarding the patient and the treatment, with the difference being that the medical history was not taken into consideration for these patients, and also that the concomitant treatment was with NSAIDs, and not DMARDs and corticosteroids.

The problem had the following input attributes: age (integer value, in years), gender (binary value), biological therapy (symbolic, with the following types: 1 = BENEPALI, 2 = COSENTYX, 3 = ENBREL, 4 = HUMIRA, 5 = REMICADE, 6 = SIMPONI), the difference from the beginning of therapy (integer value, in days), and the concomitant use of NSAIDs (binary value). The output was also binary and consisted of the QuantiFERON-TB Gold Test indicator, where 1 suggested the presence of tuberculosis, and 0 the absence of the disease.

Data preprocessing regarding the duration of the therapy was conducted for the RA database as the difference of days between the date of the medical tests and the starting date of the biological therapy.

In contrast to the RA database, this database included only one record for each patient. In this way, each patient had an equal weight in the development of the model. Table 16 presents the results obtained with this approach.

**Table 16.** Classification results for all records in the dataset.

| Classification Method | Accuracy on the Training Set | Accuracy for Cross-Validation | Accuracy for Leave-One-Out |
|---|---|---|---|
| Random forest, 100 trees | 100% | 69.8413% | 71.4286% |
| Random forest, 1000 trees | 100% | 69.8413% | 71.4286% |
| NN | 100% | 65.0794% | 63.4921% |
| kNN, k = 10, w = 1/d | 100% | 65.0794% | 66.6667% |
| C4.5, Unpruned | 82.5397% | 74.6032% | 77.7778% |
| C4.5, Pruned | 76.1905% | 76.1905% | 76.1905% |
| SVM, Puk kernel | 79.3651% | 71.4286% | 73.0159% |

This dataset is also unbalanced, with 48 instances in class 0 and 15 instances in class 1. Therefore, in order to assess the precision and recall measures, the same approach as in Section 5.2.1 was performed for data augmentation. Table 17 presents the results for the best model (random forest) for the two cases: the original dataset, and the augmented dataset. It is clear that the results are better in the latter case.

**Table 17.** Precision and recall analysis for the original and augmented datasets.

| Classification Method/Dataset | TP Rate | FP Rate | Precision | Recall | F1 |
|---|---|---|---|---|---|
| Random forest, 1000 trees, original dataset | 0.813 | 0.667 | 0.796 | 0.813 | 0.804 |
| | 0.333 | 0.188 | 0.357 | 0.333 | 0.345 |
| | 0.698 | 0.553 | 0.691 | 0.698 | 0.695 |
| Random forest, 1000 trees, augmented dataset | 0.708 | 0 | 1 | 0.708 | 0.829 |
| | 1 | 0.292 | 0.774 | 1 | 0.873 |
| | 0.854 | 0.146 | 0.887 | 0.854 | 0.851 |

Another analysis referred to the evolution of the positive patients, for which the QuantiFERON-TB Gold Test indicator was initially 0 and, at one point, became 1. In this case, we wanted to highlight a symbolic model for this medical situation. Beyond the numerical results, the aim was to provide the medical staff with some explanations for the conditions that could change the health of patients in the sense mentioned above.

Table 18 shows the results obtained with the NNGE (non-nested generalized exemplars) algorithm, which is based on an instance-based learning paradigm, but determines generalized specimens, i.e., hyper-rectangles in the problem space, consisting of a set of individual instances that belong to the same class and cannot contain instances of a different class. These hyper-rectangles can be interpreted as rules. Also, the C4.5 algorithm is applied again to determine a decision tree (Table 19).

**Table 18.** Example of rules for QuantiFERON prediction, obtained with the NNGE algorithm.

```
Class 0
IF: 31.0 <= age <= 66.0 ^ gender in {0,1} ^ bio_therapy in {4} ^ 1442.0 <= diff_beginning <= 3346.0 ^ NSAIDs in {0,1} (13)
IF: 23.0 <= age <=6 5.0 ^ gender in {0,1} ^ bio_therapy in {2,4} ^ 223.0 <= diff_beginning <= 623.0 ^ NSAIDs in {0,1} (6)
IF: 26.0 <= age <= 48.0 ^ gender in {0,1} ^ bio_therapy in {4,6} ^ 698.0 <= diff_beginning <= 1050.0 ^ NSAIDs in {0,1} (6)
IF: 33.0 <= age <= 64.0 ^ gender in {0,1} ^ bio_therapy in {5} ^ 3164.0 <= diff_beginning <= 3928.0 ^ NSAIDs in {0} (4)
IF: 49.0 <= age <= 72.0 ^ gender in {1} ^ bio_therapy in {4,5,6} ^ 1140.0 <= diff_beginning <= 1367.0 ^ NSAIDs in {0} (3)
IF: 58.0 <= age <= 62.0 ^ gender in {0,1} ^ bio_therapy in {3} ^ 1023.0 <= diff_beginning <= 1431.0 ^ NSAIDs in {0,1} (3)
IF: 33.0 <= age <= 53.0 ^ gender in {1} ^ bio_therapy in {3} ^ 1849.0 <= diff_beginning <= 2227.0 ^ NSAIDs in {0} (3)
IF: 43.0 <= age <= 67.0 ^ gender in {1} ^ bio_therapy in {1} ^ 384.0 <= diff_beginning <= 445.0 ^ NSAIDs in {0,1} (2)
IF: age = 36.0 ^ gender in {0} ^ bio_therapy in {6} ^ diff_beginning = 424.0 ^ NSAIDs in {0} (1)
IF: age = 70.0 ^ gender in {1} ^ bio_therapy in {5} ^ diff_beginning = 2661.0 ^ NSAIDs in {0} (1)
IF: age = 27.0 ^ gender in {1} ^ bio_therapy in {4} ^ diff_beginning = 2342.0 ^ NSAIDs in {0} (1)
IF: age = 34.0 ^ gender in {1} ^ bio_therapy in {3} ^ diff_beginning = 584.0 ^ NSAIDs in {0} (1)
IF: age = 79.0 ^ gender in {1} ^ bio_therapy in {3} ^ diff_beginning = 4881.0 ^ NSAIDs in {0} (1)
IF: age = 46.0 ^ gender in {0} ^ bio_therapy in {3} ^ diff_beginning = 2478.0 ^ NSAIDs in {0} (1)
IF: age = 47.0 ^ gender in {1} ^ bio_therapy in {3} ^ diff_beginning = 4532.0 ^ NSAIDs in {0} (1)
IF: age = 41.0 ^ gender in {1} ^ bio_therapy in {3} ^ diff_beginning = 3556.0 ^ NSAIDs in {0} (1)

Class 1
IF: 53.0 <= age <= 60.0 ^ gender in {1} ^ bio_therapy in {3} ^ 2282.0 <= diff_beginning <= 2893.0 ^ NSAIDs in {0} (3)
IF: age = 47.0 ^ gender in {0,1} ^ bio_therapy in {4} ^ 1107.0 <= diff_beginning<=1386.0 ^ NSAIDs in {0} (2)
IF: 45.0 <= age <= 65.0 ^ gender in {0,1} ^ bio_therapy in {3} ^ 322.0 <= diff_beginning <= 660.0 ^ NSAIDs in {0} (2)
IF: 38.0 <= age <= 40.0 ^ gender in {1} ^ bio_therapy in {1,3} ^ 427.0 <= diff_beginning <= 1526.0 ^ NSAIDs in {0} (2)
IF: 35.0 <= age <= 39.0 ^ gender in {1} ^ bio_therapy in {5} ^ 47.0<=diff_beginning<=1224.0 ^ NSAIDs in {0} (2)
IF: 63.0 <= age <= 67.0 ^ gender in {1} ^ bio_therapy in {5} ^ 2185.0 <= diff_beginning <= 2568.0 ^ NSAIDs in {0} (2)
IF: age = 33.0 ^ gender in {0} ^ bio_therapy in {6} ^ diff_beginning = 511.0 ^ NSAIDs in {0} (1)
IF: age = 30.0 ^ gender in {1} ^ bio_therapy in {5} ^ diff_beginning = 2393.0 ^ NSAIDs in {0} (1)
```

**Table 19.** Decision tree for QuantiFERON prediction, obtained with the unpruned C4.5 algorithm on the training set.

```
NSAIDs = 0
| bio_therapy = 1: 0 (2.0/1.0)
| bio_therapy = 2: 0 (1.0)
| bio_therapy = 3: 0 (16.0/6.0)
| bio_therapy = 4: 0 (19.0/2.0)
| bio_therapy = 5
| | diff_beginning <= 2591: 1 (6.0/1.0)
| | diff_beginning > 2591: 0 (5.0)
| bio_therapy = 6: 0 (6.0/1.0)
NSAIDs = 1: 0 (8.0)
```

The final analysis was based on the application of attribute selection algorithms to confirm the propositions resulting from the experience of the medical expert. Table 20 shows

the results provided by three specialized algorithms, which indicate that the attributes considered above are relevant to our problem.

**Table 20.** The results of the attribute selection algorithms.

| InfoGain | | ReliefF | | CFS |
|---|---|---|---|---|
| 0.1026 | bio_therapy | 0.2175 | bio_therapy | bio_therapy |
| 0.05385 | NSAIDs | 0.046 | NSAIDs | NSAIDs |
| 0.00185 | gender | 0.0365 | gender | |
| 0 | diff_beginning | 0.0182 | diff_beginning | |
| 0 | age | −0.0127 | age | |

For the AS patients, all the selection algorithms concluded that the biological therapy is the most important factor for the risk of TB, followed by NSAID use and patients' genders. The use of bDMARDs in AS patients increases the risk of TB, especially for those treated with TNF inhibitors. It is also known that there is a substantially larger incidence of TB in men than in women, so patients' genders play an important role. The research data regarding the risk of TB related to NSAID use is contradictory.

## 6. Conclusions

The present approach aims to analyze the incidence of latent tuberculosis in patients with rheumatoid arthritis (76 patients) and ankylosing spondylitis (63 patients) treated with biological disease-modifying antirheumatic drugs (bDMARDs), depending on the biological agent and the treatment with conventional DMARDs (csDMARDs) and nonsteroidal anti-inflammatory drugs (NSAIDs). Consequently, the study is divided into two problems.

Regarding the patients with rheumatoid arthritis, to goal of this part of the study was to identify the reactivation of latent tuberculosis, evidenced by the positive QuantiFERON test result in patients undergoing biological therapy. We aimed to establish the factors that influence the reactivation of tuberculosis, taking into account both elements related to the patient (age, gender, and medical history) and the treatment they followed (the type of biological therapy, the use of conventional DMARDs or corticosteroids, and the duration of therapy), in order to determine which of them are relevant for our problem.

For classification, the classical algorithms used generally gave good results with the implementation in WEKA, a popular collection of machine learning algorithms: random forest, nearest neighbor, k-nearest neighbors, non-nested generalized exemplar, C4.5 decision trees, and support vector machines.

The results obtained are presented in two variants: on the entire training set, to highlight the ability of a model to learn the data, and with a 10-fold cross-validation, to highlight the generalization capability of the learned model.

Two of the attribute selection algorithms (ReliefF, InfoGain, and correlation-based feature selection) concluded that the biological therapy (bDMARDs) and some of the csDMARDs (Methotrexate, Hydroxychloroquine, Sulfasalazine, and Leflunomide), play an important role in developing latent TB in the RA patients treated with biological therapy. These results are consistent with information from medical studies, as it is known that biologic therapy involves an increased risk of the reactivation of latent TB in patients with rheumatoid arthritis. The risk of mycobacterial infection is also increased in RA patients treated with csDMARDs, but the risk is lower than in those receiving bDMARDs.

The third attribute selection algorithm (ReliefF) concluded that three of the most frequently used csDMARDs (Leflunomide, Methotrexate, and Hydroxychloroquine) play the most important role in developing LTB. This algorithm also considered that the gender of the patient is important.

For the ankylosing spondylitis database, which was the second part of the study, the aim of the modelling was also to identify the reactivation of latent tuberculosis in patients undergoing biological therapy by considering the positive result of the QuantiFERON test. The relevant factors for our problem were also selected from elements regarding the

patient and the treatment, with the difference being that medical history was not taken into consideration for these patients, and that the concomitant treatment was with NSAIDs, and not DMARDs and corticosteroids.

For the AS patients, all the selection algorithms concluded that the biological therapy is the most important factor for the risk of TB, followed by NSAID use and the patients' genders. The use of bDMARDs in AS patients increases the risk of TB, especially for those treated with TNF inhibitors. It is also known that there is a substantially larger incidence of TB in men than in women, so patients' genders play an important role. The research data regarding the risk of TB related to NSAID use is contradictory.

The importance of this analysis lies in the fact that the medical problem has a practical utility, and the training data belongs to real patients. In addition, beyond the results obtained from a qualitative point of view (the accuracy of the predictions), the applied methodology is an important tool for a study by simulation that has prediction opportunities.

**Author Contributions:** Conceptualization: S.C. and F.L.; methodology: S.C., F.L., A.-M.M.-V. and E.R.; software: F.L.; validation: S.C., A.-M.M.-V. and E.R.; formal analysis: A.-M.M.-V. and E.R.; investigation: S.C., A.-M.M.-V. and E.R.; resources: S.C., F.L., A.-M.M.-V. and E.R.; writing—original draft preparation, S.C., F.L. and A.-M.M.-V.; writing—review and editing, S.C., F.L. and A.-M.M.-V.; supervision: S.C. and E.R.; funding acquisition: S.C. All authors have read and agreed to the published version of the manuscript.

**Funding:** This work was supported by Exploratory Research Projects PN-III-P4-ID-PCE-2020-0551, no. 91/2021, financed by UEFISCDI.

**Informed Consent Statement:** All authors have read and agreed to the published version of the manuscript.

**Data Availability Statement:** The data were collected by the authors according to the available normative.

**Acknowledgments:** This work was supported by Exploratory Research Projects PN-III-P4-ID-PCE-2020-0551, no. 91/2021, financed by UEFISCDI.

**Conflicts of Interest:** The authors declare no conflict of interest.

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
