# Peer review of "Analyzing Tuberculosis Reactivation in Patients with Rheumatoid Arthritis and Ankylosing Spondylitis Treated with Biological Therapy Using Machine Learning Methods"

_applsci, doi:10.3390/app112311400_

Round 1
Reviewer 1 Report
In the medical field, machine learning research should provide solutions to clinically important problems. The issues of this study is clinically meaningful and relevant. But there are a few drawbacks to solve this issue. First, output variables should be reactivation of TB not quantiFERON gold. The reactivation of TB and positive quantiFERON test is quite different. The quantiFERON test is very simple. I cannot find any clinical significance to predict the result of quantiFERON gold test. For this study, data of patients who had reactivation of TB should be collected. Second, it is difficult to find any clinical implication or lessons. The researches should collect various clinical features of RA and AS patients, and provide important clinical features to predict reactivation of TB as conclusion. But there is no data about patient’s clinical features or characteristics. Simple comparison of prediction model performance is not helpful for clinical practice. In addition, introduction is quite long and tedious. It should be concise and simple. In my opinion, the researchers need to communicate with clinicians. If there is a synergistic effect between researchers and clinicians, a better result will be produced.
Author Response
In the attached file you will find the answers to the observations .

Reviewer 2 Report
The authors in this paper present the use of machine learning methods to detect latent tuberculosis in patients with rheumatoid arthritis and ankylosing spondylitis treated with biological disease-modifying antirheumatic drugs.
The article describes the medical background very well and accurately shows the dataset used.
The article from a medical point of view can be pretty interesting. However, there are quite a few problems from a computer science and machine learning point of view.
Basically, the artificial intelligence methods presented have been known for decades and do not present any new value.
Furthermore, the dataset used is dramatically tiny (less than 100 observations) and very strongly unbalanced.
Due to the strong unbalance of the set, presenting accuracy is entirely unjustified. The authors should focus on the analysis of the detection of all true positives.
Moreover, performing 10-fold cross-validation on such a small set leads to completely distorted results. A better solution would be to use leave-one-out.
Tables 9-12 and 15-16 are nice trivia but do not contribute anything to the paper.
The numerical experiment was conducted on such a small sample that it is impossible to analyze statistical significance, so data dimensionality reduction from a machine learning point of view is unfortunately also unjustified.
Author Response
In the attached file you will find the answers to the observations

Round 2
Reviewer 1 Report
No comments
Reviewer 2 Report
The authors in this paper present the use of machine learning methods to detect latent tuberculosis in patients with rheumatoid arthritis and ankylosing spondylitis treated with biological disease-modifying antirheumatic drugs.
The article describes the medical background very well and accurately shows the dataset used.
All my comments and remarks were satisfactorily addressed.